# Do Hungarian multiple sclerosis care units fulfil international criteria?

**Zsófia Kokas[1], Dániel Sandi[1], Zsanett Fricska-Nagy[1], Judit Füvesi[1], Tamás Biernacki[1], Ágnes Köves[2], Ferenc Fazekas[3], Adrienne Jóri Birkás[4], Gabriella Katona[5], Krisztina Kovács[6], Dániel Milanovich[7], Enikő Dobos[8], István Kapás[9], Gábor Jakab[10], Tünde Csépány[11], Erzsébet Bense[12], Klotild Mátyás[13], Gábor Rum[14], Zoltán Szolnoki[15], István Deme[16], Zita Jobbágy[17], Dávid Kriston[18], Zsuzsanna Gerócs[19], Péter Diószeghy[20], László Bors[21], Adrián Varga[22], Levente Kerényi[23], Gabriella Molnár[24], Piroska Kristóf[25], Zsuzsanna Ágnes Nagy[26], Mária Sátori[27], Piroska Imre[28], Szilvia Péntek[29], Péter Klivényi[1], Zsigmond Tamás Kincses[1,30], László Vécsei[1,31], Krisztina Bencsik[1]** * 

1 Faculty of General Medicine, Department of Neurology, Albert Szent-Györgyi Clinical Centre, University of Szeged, Szeged, Hungary, 2 Department of Neurology, Bajcsy-Zsilinszky Hospital, Budapest, Hungary, 3 Department of Neurology, Gyula Nyírő Hospital and National Institute of Psychiatry and Addictions, Budapest, Hungary, 4 Department of Neurology, National Institute of Clinical Nerosciences, Budapest, Hungary, 5 Department of Neurology, National Institute of Rheumatology and Physiotherapy, Budapest, Hungary, 6 Department of Neurology, Péterfy Hospital, Budapest, Hungary, 7 Department of Neurology, Semmelweis University, Budapest, Hungary, 8 Department of Neurology, Saint Imre Hospital and University Teaching Hospital, Budapest, Hungary, 9 Department of Neurology, Saint János Hospital, Budapest, Hungary, 10 Department of Neurology, Uzsoki Hospital, Budapest, Hungary, 11 Division of Neurology, University of Debrecen Clinical Center, Debrecen, Hungary, 12 Department of Neurology, University of Debrecen Faculty of Medicine, Debrecen, Hungary, 13 Department of Neurology, Ferenc Markhot Teaching Hospital, Eger, Hungary, 14 Department of Neurology, Aladár Petz University Teaching Hospital, Győr, Hungary, 15 Department of Neurology, Kálmán Pándy County Hospital, Gyula, Hungary, 16 Department of Neuology, Mór Kaposi Teaching Hospital, Kaposvár, Hungary, 17 Department of Neurology, Kecskemét County Hospital, Kecskemét, Hungary, 18 Department of Neurology, Borsod-Abaúj-Zemplén County Central Hospital and University Teaching Hospital, Miskolc, Hungary, 19 Department of Neurology, Dorottya Kanizsai Hospital, Nagykanizsa, Hungary, 20 Department of Neurology, Aladár Jósa Teaching Hospital, Nyíregyháza, Hungary, 21 Department of Neurology, University of Pécs Clinical Center Pécs, Pécs, Hungary, 22 Department of Neurology, Saint Lázár County Hospital, Salgótarján, Hungary, 23 Department of Neurology, Fejér County Saint György University Teaching Hospital, Székesfehérvár, Hungary, 24 Department of Neurology, János Balassa Hospital, Szekszárd, Hungary, 25 Department of Neurology, Jász-Nagykun-Szolnok County Géza Hetényi Hospital, Szolnok, Hungary, 26 Department of Neurology, Markusovszky University Teaching Hospital, Szombathely, Hungary, 27 Department of Neurology, Saint Borbála Hospital, Tatabánya, Hungary, 28 Department of Neurology, Ferenc Csolnoky Hospital, Veszprém, Hungary, 29 Department of Neurology, Zala County Saint Rafael Hospital, Zalaegerszeg, Hungary, 30 Faculty of General Medicine, Department of Radiology, Albert Szent-Györgyi Clinical Centre, University of Szeged, Szeged, Hungary, 31 MTA-SZTE Neuroscience Research Group, Szeged, Hungary

* bencsik.krisztina@med.u-szeged.hu

## Abstract

### A patients

Because of the past 3 decades' extensive research, several disease modifying therapies became available, thus a paradigm change is multiple sclerosis care was necessary. In 2018 a therapeutic guideline was created recommending that treatment of persons with multiple sclerosis should take place in specified care units where the entire spectrum of disease modifying therapies is available, patient monitoring is ensured, and therapy side effects are detected and treated promptly. In 2019 multiple sclerosis care unit criteria were developed,

**Data Availability Statement:** All relevant data are within the paper and its Supporting Information files.

**Funding:** This study was funded via the University of Szeged in the form of independent grants to ZsK

(5535 / EFOP 3.6.3-VEKOP-16-2017-00009) and LV (GINOP 2.3.2-15-2016-00034 / TUDFO/47138-1/2019-ITM). The funders had no role in study design, data collection, and analysis, decision to publish, or preparation of the manuscript. No additional external funding was received for this study.

**Competing interests:** The authors have declared that no competing interests exist.

emphasizing personnel and instrumental requirements to provide most professional care. However, no survey was conducted assessing the real-world adaptation of these criteria.

## Objective

To assess whether Hungarian care units fulfil international criteria.

## Methods

A self-report questionnaire was assembled based on international guidelines and sent to Hungarian care units focusing on 3 main aspects: personnel and instrumental background, disease-modifying therapy use, number of people living with multiple sclerosis receiving care in care units. Data on number of persons with multiple sclerosis were compared to Hungarian prevalence estimates. Descriptive statistics were used to analyse data.

## Results

Out of 27 respondent care units, 3 fulfilled minimum requirements and 7 fulfilled minimum and recommended requirements. The least prevalent neighbouring specialties were spasticity and pain specialist, and neuro-ophthalmologist and oto-neurologist. Only 15 centres used all available disease modifying therapies. A total number of 7213 people with multiple sclerosis received care in 27 respondent centres. Compared to prevalence estimates, 2500 persons with multiple sclerosis did not receive multiple sclerosis specific care in Hungary.

## Conclusion

Less than half of Hungarian care units provided sufficient care for people living with multiple sclerosis. Care units employing fewer neighbouring specialties, might have difficulties diagnosing and providing appropriate care for persons with multiple sclerosis, especially for people with progressive disease course, contributing to the reported low number of persons living with multiple sclerosis.

## 1. Introduction

### 1.1. Background

Multiple Sclerosis (MS) is an autoimmune demyelinating neurodegenerative disorder of the central nervous system (CNS), which usually affects young adults [1]. Although MS is a rare disease, it affects roughly 2.8 million people worldwide [2] and is one of the most cost-expensive diseases to treat nowadays [3]. Without appropriate treatment, people with MS develop physical and cognitive impairment, as well as other psychopathological anomalies, such as depression, anxiety and fatigue, which all significantly diminish working ability, societal relationships and quality of life [4].

### 1.2. First therapy and its effects on the management of persons living with multiple sclerosis

The very first therapy approved in the treatment of MS was interferon-β-1b (IFN-β-1b) [5], which was licensed in the European Union (EU) in 1995. In the EU, persons with MS were

prescribed IFN-β-1b by local neurologists and were provided with their medication according to the funding protocol of national insurance companies. After prescribing the disease modifying therapy (DMT), management of people living with MS was provided by general practitioners (GP) and in some cases care of persons living with MS took place in University Departments to participate in clinical research. Because of the differences in resources, health care system and legal background among countries, MS care could not have been standardized, and since then vast differences occurred in management of people with MS.

## 1.3. Management of people living with multiple sclerosis in Hungary

DMT prescribing in Hungary was approached differently because of the high retail price of the disease modifying therapies. Disease specific therapy prescription was allowed by the National Health Insurance Fund (NHIF) if MS care was pursued in specialized MS centres. Thus, specialized MS centre conditions had to be determined by the Hungarian Neurological Professional College in 1996 (Table 1).

In that guideline besides professional criteria, patient's equality was also taken into consideration. In this manner, the four Hungarian University Departments, county hospitals and some regular city hospitals in the capital city (26 in total), were designated to participate in MS care. Since then, this number steadily increased to 31 (Fig 1). In the past 25 years Hungarian MS centres have operated according to Neurological Professional College's guideline. However, in this period classification of disease course, diagnostic criteria of MS and magnetic resonance imaging (MRI) protocols have changed multiple times, thus Hungarian MS centre conditions can be considered outdated.

## 1.4. Standardizing management of persons with multiple sclerosis

Over the past 25 years several aspects of MS have changed. As previously mentioned, disease course classification [6–8], diagnostic criteria [9–12], magnetic resonance imaging guidelines [13,14], the therapeutic arsenal and recommendations have rapidly evolved. Currently, due to the extensive research of the last three decades, all courses of the disease can be treated with various DMTs and symptomatic treatments [15]. Due to the rapid evolution of MS knowledge, it is comprehensible that GPs and general neurologists without experience in the MS field are not able to keep abreast of brain-health focused care [16], thus there was an increasing need for standardizing MS clinical care. Therefore, the latest therapeutic guideline was developed by the European Committee for Treatment and Research in Multiple Sclerosis with the contribution of European Academy of Neurology (ECTRIMS/EAN) [17,18] in 2018. The first recommendation of that guideline was that DMTs should only be prescribed in multiple sclerosis

**Table 1. Specialized MS centre conditions determined by the Hungarian Neurological Professional College.**

| Hungarian MS centre conditions |
| --- |
| 1. The neurological department should have a separate outpatient unit dedicated only for persons with MS, with at least 6 consulting hours a week. |
| 2. At least 2 neurologists with experience in treating MS and at least 1 specially trained MS nurse should be employed. |
| 3. The centre should have the conditions to properly examine, diagnose, and treat people with MS. |
| 4. Documentation should include patient history, annual relapse rate change after DMT initiation, change of physical status (Expanded Disability Status Scale) while receiving DMT |
| 5. Eligibility to treatment was approved by an independent board of professionals thoroughly reviewing disease history recorded in documentation. |

Abbreviations: DMT = disease modifying therapy, MS = multiple sclerosis.

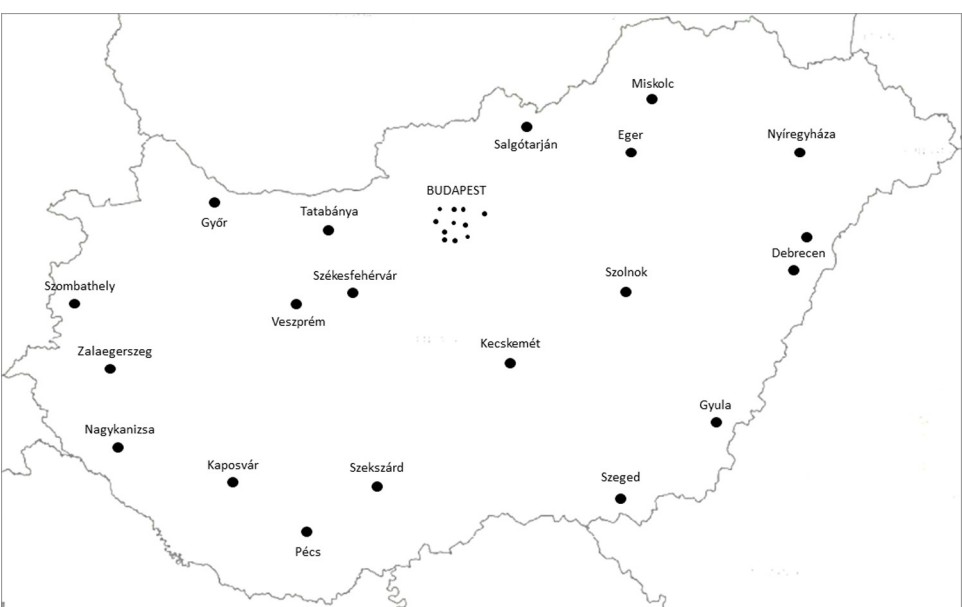

**Fig 1. Location of Hungarian MS care units.** Black dots indicate the location of Hungarian MS care units. This map clearly represents that MS care unit density is highest in Pest County, where the capital city of Budapest with highest population density is located, whereas in other counties with less residents, usually 1 care unit/county is responsible for MS patient management. Abbreviations: MS = multiple sclerosis.

care units (MSCU), since MSCUs have the appropriate infrastructure to ensure adequate administration of DMTs, treatment if adverse reactions occur and proper monitoring of people with MS and treatment response [17,18]. Subsequently, in 2019 MS care unit criteria [19], were also developed, in which personnel, instrumental, associate professional requirements of care units were defined (Table 2).

## 1.5. Objectives

Since the creation of these guidelines [17–19], no survey was conducted to assess the real-world adaptation of these criteria. Hence, in 2019 the Danube Symposium for Neurological Sciences (DSNS) member countries decided to create the Multiple Sclerosis National Symposium, aiming to collect and assess data regarding inpatient and outpatient care of people with MS.

The present study is part of an international survey series conducted in DSNS member countries. Taking into account that such robust data collection is time and human resource

**Table 2. Multiple sclerosis care unit recommendations.**

| Summary of the Multiple Sclerosis Care Unit criteria |
| --- |
| 1. Core of the MSCU is the person living with MS, MS neurologist, MS nurse and at least 3 of the following: pharmacist, neuropsychologist, speech therapist, dietitian, continence specialist, pain specialist, spasticity specialist (the latter 3 can be provided by the MS nurse or the MS specialist). <br> 2. To ensure accurate diagnostics and differential diagnostics a qualified neuro-radiologist, microbiology, laboratory, electrophysiology and ophthalmology and an MRI device should also be accessible. <br> 3. A fully developed MSCU should collaborate with many other neighbouring specialties as well: neuro-ophthalmologist, oto-neurologist, neurorehabilitation, neurosurgeon, surgeon, obstetrician-gynaecologist, internal medicine specialist, psychiatrist to treat complications and manage comorbidities. |

Abbreviations: MS = multiple sclerosis, MSCU = multiple sclerosis care unit.

demanding, data acquisition and analysis from DSNS member countries is in process. At the same time, considering the lack of similar assessments in the past, the present study can be considered as a pilot investigation aiming to collect data only from Hungarian MS care units, before presenting international outcomes. The main objective was to assess whether Hungarian MS care units fulfil international criteria. This objective could be divided into three subobjectives:

1. to assess Hungarian MS care units' personnel and infrastructural equipment,

2. to enquire about DMT use

3. to determine number of persons living with MS receiving care in these units, and to compare these data to national prevalence estimates.

## 2. Materials and methods

### 2.1. Study design

The survey was conducted at the Department of Neurology University of Szeged, Albert Szent-Györgyi Health Centre, Szeged, Hungary with the contribution of Hungarian MS care units. The main goal of the survey was to assess personnel and infrastructural criteria of MS care units. Hence, a self-report questionnaire was assembled according to the ECTRIMS/EAN [17,18] and the MS care unit [19] guidelines at the beginning of November, 2020, and the end-date of data collection was January 31, 2021. The questionnaire consisted of 3 main sections: first section assessing personnel and instrumental equipment, second section examining DMTs used in the care unit, and the third section surveying number of people with MS receiving care at the MSCU (Tables 3 and 4). Resources on which the questionnaire was based, were attached to the main file, to minimize misunderstanding amongst colleagues. NHIF data were also gathered regarding DMT use of December 2020 to get a comprehensive overview on proportion of moderately and highly active therapy use in a national level.

### 2.2. Ethical approval

The study was approved by the Hungarian Medical Research Council, reference number: IV/5139-1/2021/EKU. The study was conducted in accordance with the Declaration of Helsinki.

### 2.3. Statistical analysis

Descriptive statistics were used to summarize data.

## 3. Results

### 3.1. Participation rate

The total number of respondents were 29 out of 31 care units, resulting in a 93.55% participation rate. However, only 24 care units provided information to every aspect of all sections of the questionnaire (Table 5). Yet, all 29 received questionnaires were included in the analysis, irrespective of being filled in partially or completely. To ensure anonymity, care units participating in the assessment were numbered consecutively from 1 to 29.

### 3.2. Care unit personnel and instrumental background

Regarding number of MS neurologist, 26 care units provided information, giving a total number of 86. The median number of MS neurologists was 3 (range, 1–8). When examining the

**Table 3. MS care unit criteria–first part of the questionnaire.**

| MS care unit criteria–detailed questionnaire | |
|---|---|
| **Minimum requirements of a multidisciplinary MS care unit** | **Occupied in the MS care unit** |
| **Core of the MS care unit** | |
| Number of persons with MS receiving care | _________ |
| Number of MS neurologists | _________ |
| MS nurse | ☐ Yes / ☐ No |
| Secretary | ☐ Yes / ☐ No |
| **Collaboration with part-time specialists** | |
| Neuropsychologist | ☐ Yes / ☐ No |
| Pharmacist with special knowledge of DMTs | ☐ Yes / ☐ No |
| Dietitian | ☐ Yes / ☐ No |
| Speech therapist | ☐ Yes / ☐ No |
| Pain specialist | ☐ Yes / ☐ No |
| Continence specialist | ☐ Yes / ☐ No |
| Spasticity specialist | ☐ Yes / ☐ No |
| **Recommended requirements to achieve a fully developed multidisciplinary MS care unit** | **Occupied in the MS care unit** |
| **Collaboration with other specialties** | |
| Radiology with MS-familiar neuro-radiologist | ☐ Yes / ☐ No |
| Physician | ☐ Yes / ☐ No |
| Surgeon | ☐ Yes / ☐ No |
| Neurosurgeon | ☐ Yes / ☐ No |
| Obstetrician gynaecologist | ☐ Yes / ☐ No |
| Neuro-ophthalmologist | ☐ Yes / ☐ No |
| Neuro-otologist | ☐ Yes / ☐ No |
| Psychiatrist | ☐ Yes / ☐ No |
| Neurorehabilitation | ☐ Yes / ☐ No |

Abbreviations: MS = multiple sclerosis.

rest of the MS care unit minimum and recommended requirements 27/29 care units provided information regarding personnel and infrastructural background. From the respondent care units 26 employed an MS nurse, and 21 employed a secretary. Concerning collaboration with other specialties, spasticity specialist (13/29), pain specialist (15/29), neuro-ophthalmologist (15/29) and oto-neurologist (15/29), neuropsychologist (19/29), speech therapist (21/29) were the least prevalent. Other required specialty missing was MS radiologist, absent in 3 care units, in 5 care units neurorehabilitation was not accessible and 2 care units did not own an MRI

**Table 4. MS care unit criteria–second part of the questionnaire.**

| Currently used DMTs in the MS care unit | | | |
|---|---|---|---|
| For low disease activity | For high disease activity | For very high disease activity | Other |
| ☐ Interferon β | ☐ Natalizumab | ☐ Alemtuzumab | ☐ Mitoxantrone |
| ☐ Glatiramer acetate | ☐ Fingolimod | ☐ Ocrelizumab | ☐ Azathioprine |
| ☐ Dimethyl fumarate | | ☐ Cladribine | ☐ Cyclophosphamide |
| ☐ Teriflunomide | | | ☐ Siponimod |

Abbreviations: DMT = disease modifying therapy.

**Table 5. Proportion of care units providing information on different aspects of the questionnaire.**

| Aspects | Proportion of MSCUs providing information |
|---|---|
| Personnel and instrumental background of the MSCU: Number of MS neurologist | 26/29 (89.66%) |
| Personnel and instrumental background of the MSCU: MS nurse and secretary employment, neighbouring specialties, and diagnostics | 27/29 (93.10%) |
| DMT use | 27/29 (93.10%) |
| Number of MS patients | 27/29 (93.10%) |

Abbreviations: DMT = disease modifying therapy, MS = multiple sclerosis, MSCU = multiple sclerosis care unit.

device. Summarizing the data 3/29 care units fulfilled only the minimum, 7/29 centres fulfilled both the minimum and recommended, while the remaining 17 centres did not fulfil all aspects of either minimum or recommended criteria and 2 care units did not provide information on personnel and infrastructural background (Fig 2).

## 3.3. Disease modifying therapy use in care units

Concerning DMT use 27/29 care units provided information. In 27/29 care units all low efficacy DMTs were administered. Considering highly effective oral and infusion therapies there

| Organization of the MS care units | | | 1. | 2. | 3. | 4. | 5. | 6. | 7. | 8. | 9. | 10. | 11. | 12. | 13. | 14. | 15. | 16. | 17. | 18. | 19. | 20. | 21. | 22. | 23. | 24. | 25. | 26. | 27. | 28. | 29. | Total | % |
|---|---|---|---|---|---|---|---|---|---|---|---|---|---|---|---|---|---|---|---|---|---|---|---|---|---|---|---|---|---|---|---|---|---|
| Minimum requirements of a multidisciplinary MS care unit | Core | Number of persons with MS | 150 | 92 | 320 | N/A | 400 | 500 | 269 | 40 | 415 | 400 | 119 | 200 | 400 | 240 | 196 | 200 | 400 | 53 | 250 | 411 | 88 | 950 | 120 | 70 | 122 | 250 | 348 | 210 | N/A | 7213 | N/A N/A |
| | | Number of MS neurologists | 5 | 2 | 4 | N/A | 2 | 5 | 2 | 2 | 6 | 6 | 3 | 2 | 4 | 3 | 2 | 2 | 1 | N/A | 3 | 8 | 1 | 7 | 3 | 1 | 3 | 2 | 5 | 2 | N/A | 86 | N/A N/A |
| | | MS nurse | X | X | X | X | X | X | X | X | X | X | X | X | X | X | X | X | X | X | X | X | X | N/A | X | X | | X | N/A | X | X | X | 26/29 | 89.66% |
| | | Secretary | | X | X | | X | X | | X | X | X | X | X | X | X | | | X | X | X | X | N/A | X | X | | X | N/A | X | X | X | | 21/29 | 72.41% |
| | Collaboration with part-time specialists | Neuropsychologist | X | X | X | X | X | X | | X | X | X | X | | X | | X | | X | | X | X | N/A | X | X | | | N/A | X | X | | 19/29 | 65.52% |
| | | Pharmacist | X | X | X | X | X | X | X | X | X | X | X | X | X | X | X | X | X | X | X | X | N/A | X | X | X | X | N/A | X | X | X | 27/29 | 93.10% |
| | | Dietitian | X | X | X | X | X | X | X | X | X | X | X | X | X | X | X | X | X | X | X | X | N/A | X | X | X | X | N/A | X | X | X | 27/29 | 93.10% |
| | | Speech therapist | X | | X | | X | X | X | X | X | X | X | X | X | | X | | X | | X | X | N/A | X | X | X | X | | | | | 21/29 | 72.41% |
| | | Pain specialis | | X | X | X | X | X | | | X | X | X | | | | | X | | | X | X | N/A | X | X | | | N/A | | | | 15/29 | 51.72% |
| | | Continence specialist | X | X | | X | X | X | | X | X | X | X | X | X | X | X | X | X | X | X | X | N/A | X | X | X | X | N/A | | X | X | 25/29 | 86.21% |
| | | Spasticity specialist | | X | | X | X | | | X | X | X | | | | | | | X | X | X | X | N/A | X | X | | | N/A | | X | | 13/29 | 44.83% |
| Recommended to achieve the fully developed multidisciplinary MS care unit | Collaboration with other specialties | MS-familial neuro-radiologist | X | X | X | X | X | X | X | | X | X | X | X | X | X | X | X | X | X | X | X | N/A | X | X | | | N/A | X | X | X | 24/29 | 82.76% |
| | | Microbiology | X | X | X | X | 2 | X | X | X | X | X | X | X | X | X | X | X | X | X | X | X | X | X | X | X | X | N/A | X | X | X | 27/27 | 93.10% |
| | | Laboratory | X | X | X | X | X | X | X | X | X | X | X | X | X | X | X | X | X | X | X | X | X | X | X | X | X | N/A | X | X | X | 27/27 | 93.10% |
| | | Electrophysiology | X | X | X | X | X | X | X | X | X | X | X | X | X | X | X | | X | X | X | N/A | X | X | X | X | | N/A | X | X | X | 25/27 | 86.21% |
| | | Ophtalmology | X | X | X | X | X | X | X | X | X | X | X | X | X | X | X | X | X | X | X | X | X | X | N/A | X | X | N/A | X | X | X | 27/27 | 93.10% |
| | | Physician | X | X | X | X | X | X | X | X | X | X | X | X | X | X | X | X | X | X | X | X | X | X | N/A | X | X | N/A | X | X | X | 27/29 | 93.10% |
| | | Surgeon | X | | | X | X | X | X | X | X | X | X | X | X | X | X | X | X | X | X | X | N/A | X | X | | X | N/A | X | X | X | 25/29 | 86.21% |
| | | Neurosurgeon | X | X | X | | X | X | X | X | X | X | X | X | X | X | X | X | | X | X | N/A | X | X | | X | N/A | X | X | | 23/29 | 79.31% |
| | | Obstetrician gynecologist | X | X | | | X | X | X | X | X | X | X | X | X | X | X | X | X | X | X | N/A | X | X | X | X | N/A | X | X | X | 25/29 | 86.21% |
| | | Neuro-ophtalmologist | | X | X | | X | X | X | X | X | | | | | | X | X | X | N/A | X | | | N/A | | | | 15/29 | 51.72% |
| | | Otoneurologist | X | | X | | X | | X | X | X | X | X | | | | X | X | X | N/A | X | | | N/A | | | | 15/29 | 51.72% |
| | | Psychiatrist | X | X | X | X | X | X | X | X | X | X | X | X | X | X | X | X | X | X | X | X | X | X | X | X | X | N/A | X | X | X | 27/29 | 93.10% |
| | | Neurorehabilitation | X | X | X | X | X | X | X | | X | X | X | X | X | | X | X | X | X | N/A | X | X | | X | N/A | | X | | 22/29 | 75.86% |
| Summary | | Number of missing criteria | 4 | 4 | 3 | 7 | 1 | 0 | 6 | 3 | 1 | 0 | 0 | 4 | 3 | 6 | 5 | 7 | 4 | 4 | 0 | 0 | N/A | 0 | 2 | 10 | 7 | N/A | 2 | 0 | 8 | | |
| | | Minimum criteria fulfilled | | | | | X | X | | | X | X | X | | | | | | | | X | X | N/A | X | X | | | N/A | | X | | 10/29 | 34.48% |
| | | Recommended criteria fulfilled | | | | | | X | | | X | X | | | | | | | | | X | X | N/A | X | | | | N/A | | X | | 7/29 | 24.14% |
| | | Both fulfilled | | | | | | X | | | X | X | | | | | | | | | X | X | N/A | X | | | | N/A | | X | | 7/29 | 24.14% |

**Fig 2. MS care unit criteria fulfilment among Hungarian MS care units.** Respondent Hungarian MS care units are numbered from 1 to 29. In the first column MS care unit minimum and recommended criteria are listed. In the first 2 rows patient number receiving care at the care unit and number of MS neurologists are marked. At the bottom 4 rows summary of care units fulfilling all aspects of the minimum and recommended criteria are represented as well as number of unfulfilled criteria care unit by care unit. In the last column the number of care units out of all respondent care units fulfilling a criterion is summarised. "X" indicates that criterion is fulfilled. "N/A" indicates that no data were supplied. Abbreviations: MS = multiple sclerosis.

**Table 6. Disease modifying therapy use in care units.**

|  | ß-IFN | DMF | GA | TFL | FG | CLA | NAT | ALM | OCR |
|---|---|---|---|---|---|---|---|---|---|
| **1.** | X | X | X | X | X | X | X | X | X |
| **2.** | X | X | X | X | X | X | X |  | X |
| **3.** | X | X | X | X | X | X | X | X | X |
| **4.** | N/A | N/A | N/A | N/A | N/A | N/A | N/A | N/A | N/A |
| **5.** | X | X | X | X | X | X | X | X | X |
| **6.** | X | X | X | X | X | X | X | X | X |
| **7.** | X | X | X | X | X |  | X | X | X |
| **8.** | X | X | X | X |  | X |  |  | X |
| **9.** | X | X | X | X | X | X | X | X | X |
| **10.** | X | X | X | X | X | X | X |  | X |
| **11.** | X | X | X | X | X | X | X | X | X |
| **12.** | X | X | X | X | X | X | X |  | X |
| **13.** | X | X | X | X | X | X | X | X | X |
| **14.** | X | X | X | X | X |  |  |  |  |
| **15.** | X | X | X | X | X | X | X | X | X |
| **16.** | X | X | X | X | X | X | X | X | X |
| **17.** | X | X | X | X | X | X | X | X | X |
| **18.** | X | X | X | X | X |  |  |  |  |
| **19.** | X | X | X | X | X | X | X | X | X |
| **20.** | X | X | X | X | X | X | X | X | X |
| **21.** | X | X | X | X | X |  |  |  |  |
| **22.** | X | X | X | X | X | X | X | X | X |
| **23.** | X | X | X | X | X |  | X |  | X |
| **24.** | X | X | X | X |  |  | X |  | X |
| **25.** | X | X | X | X | X | X |  | X |  |
| **26.** | N/A | N/A | N/A | N/A | N/A | N/A | N/A | N/A | N/A |
| **27.** | X | X | X | X | X | X | X | X | X |
| **28.** | X | X | X | X | X | X | X | X | X |
| **29.** | X | X | X | X | X | X | X | X |  |

In the first column respondent Hungarian Multiple sclerosis care units (numbered from 1 to 29) are listed, in the first row, abbreviations of disease modifying therapies' active substances are listed.

"X" indicates that the disease modifying therapy is used in the care units, cell with a grey background indicates that the disease modifying therapy is not used in the care unit, "N/A" means that no data were supplied.

Abbreviations: ALM = alemtuzumab, ß-IFN = ß-interferon, CLA = cladribine, DMF = dimethyl fumarate, FG = fingolimod, GA = glatiramer acetate

NAT = natalizumab, OCR = ocrelizumab, TFL = teriflunomide.

were differences among care units. In 20/29 care units all highly effective oral DMTs were used. From the remaining 7 care units 1 used only cladribine (CLA) and 5 used only fingolimod (FG) and 1 care unit did not use either FG or CLA. In 16/29 care units all highly effective infusion DMTs were used. From the remaining 11 care units 5 administered natalizumab (NAT) and ocrelizumab (OCR). In 1 care unit only NAT and alemtuzumab (ALM) was provided. In 1 care unit only OCR was administered, and in 1 care unit only ALM was used. In 3 care units neither of the highly effective infusion therapies were provided. However, there were only 15/29 care units, that administered all highly effective oral and infusion therapies, since in care unit no 7 CLA was not used, but the rest of the therapies were (Table 6).

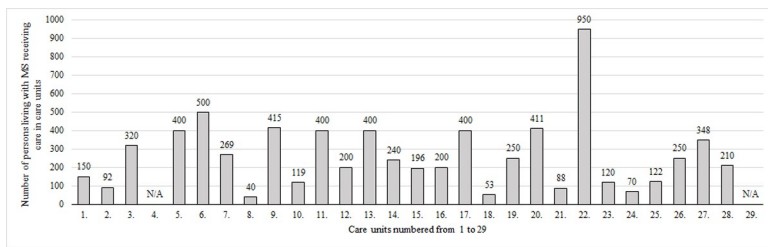

**Fig 3. Distribution of people with MS among centres.** In the linear axis respondent multiple sclerosis care units numbered from 1 to 29 are listed, in the perpendicular axis number of persons with multiple sclerosis receiving care is represented. "N/A" indicates that no data were supplied. Abbreviations: MS = multiple sclerosis.

### 3.4. Proportion of moderately and highly effective disease modifying therapy use

According to the NHIF database 4665 people with MS received DMTs in December 2020, of which 3131 (67.12%) received low efficacy DMTs. Of these persons 1360 (43.44%) used injectable treatment and 1771 (56.56%) were on oral medication. The proportion of people on highly effective therapies was lower, 1534 (32.88%) persons received either highly effective oral or infusion therapy. Of these people 810 (52.80%) used oral agents, for example cladribine or fingolimod, and 724 (47.20%) were on infusion therapy e.g., alemtuzumab, natalizumab, ocrelizumab.

### 3.5. Number of persons living with multiple sclerosis receiving care in care units

Concerning number of people with MS, 27/29 care units provided information, in those care units 7213 persons received medical care. The median number of people living with MS receiving care was 240 (range, 40–950). The 4 University Departments, 2 county hospitals and 2 general hospitals of Budapest cared for more than 400 persons. In these 8 care units 3876 (53.74%) people received medical care. In 10 care units caring for 196–348 persons, 2483 (34.42%) people received care, whereas the remaining 9 care units were accountable for the medical care of 854 (11.84%) persons living with MS (Fig 3).

## 4. Discussion

First objective of the study was to assess Hungarian MS care units' personnel and infrastructural background. According to the assessment, only 3 care units qualified for minimum, and 7 care units acquitted to both minimum and recommended requirements. On one hand, the low proportion of care units fulfilling criteria suggests that there is plenty of space for improvement for MS care units in the future. This assessment established the most crucial elements of missing criteria, thus providing opportunity to ameliorate the conditions responsible for patients not receiving equal care. On the other hand, for an ordinary MS care unit these conditions might be too strict to fulfil. The least prevalent neighbouring specialties were spasticity and pain specialist, and neuro-ophthalmologist and oto-neurologist. Since these specialties are quite rare in general, it might be worthy to consider assigning regional care units employing these specialists, providing consulting hours for less equipped centres. Another feasible solution could be to incorporate smaller care units into larger ones, considering units caring for more people with MS, usually fulfil more criteria, thus provide better care. In 7 care units either the absence of a secretary or an MS nurse was the main reason for not fulfilling minimum criteria. Yet, care units only having insufficiencies in these aspects, could quickly

increase MS nurse and secretary employment, since this is funded by the Hungarian Government.

As previously mentioned, no surveys were conducted to assess whether or how MS care unit criteria are fulfilled in real-world practice. However, these criteria were carefully reviewed from a different aspect by a panel of neurologists from Latin America [20]. In Latin America diagnosis and treatment of MS was usually carried out by general neurologists, and care of people with MS was not perused in specialized centres, thus the panel's objective was to create a realistic adaptation of the MS care unit recommendations considering regional differences regarding healthcare in Latin American countries. Cristiano et al acknowledged the importance of MSCUs to optimize MS care and reached consensus on what a care unit in Latin America should offer. However, they also concluded, that despite the desire to meet the personnel and instrumental requirements of an MSCU, it would be difficult to implement these expectations in certain regions of Latin America. Thus, they determined a standardized protocol for MRI assessments and recommended education of general neurologists to facilitate accurate diagnosis and treatment of MS. At the same time, they also proposed the idea of close collaboration between general neurologists and MSCUs, and similarly to our assessment they propounded the feasibility of reference care units.

The current assessment's second objective was to enquire about DMT use. According to the questionnaires, while all care units used the available low efficacy DMTs, only 15 care units administered all highly effective therapies as well, thus, in the rest of the care units, equality to access treatment might not be ensured. The reason behind this may be that highly effective infusion therapies might have severe side effects, such as natalizumab associated PML [21], alemtuzumab associated infusion reaction, infection and secondary autoimmune diseases [22], and ocrelizumab associated infusion reaction and infections [23]. Yet, a well-equipped MS care unit should be able to administer all DMTs and manage all side effects according to the ECTRIMS/EAN guideline [17,18]. Unlike in other countries, where not every DMT is available or funded by the health insurance, contributing to inequality [24,25], in Hungary all therapeutic options are funded by the NHIF, thus appropriate treatment of people with MS should be achievable, if adaptation to the latest guidelines came to fruition. The need for adaptation of the therapeutic guidelines was affirmed by other papers as well [26,27].

According to the NHIF database 4665 persons with MS received DMT in December 2020. However, the proportion of people with MS on DMT could not be established, since 4 care units did not provide data regarding the number of persons living with MS. Another deduction according to the NHIF data is that the use of oral DMTs is preferred in Hungary. Having said that, the proportion of people with MS on oral platform therapies could be higher, since administration is easier. On the flip side, the proportion of persons with MS on highly effective oral DMTs could be lower, since it might be assumed that, alemtuzumab and ocrelizumab decrease annual relapse rate and disease progression more effectively than fingolimod [28], even though head-to-head studies were not conducted. DMT use on a national level also suggests that in terms of therapeutic approach, escalation might still be favoured over induction. This approach may result in higher Expanded Disability Status Scale (EDSS) scores and disability, as well as lower quality of life eventually [29]. Furthermore, disability, fatigue and depression also result in lower working capacity, and unemployment raising indirect costs of MS [30].

Third objective of this survey was to determine number of people living with MS receiving care in Hungarian care units. According to our data collected from self-report questionnaires, 7213 persons with MS received medical care in 27/31 MS care units, from the remaining 4 care units 2 care units did not provide information regarding the number of people with MS, and 2 care units did not participate in our assessment at all. Considering proportion of participants,

the assessment can be considered representative, since it covers approximately 90% of the Hungarian population living with MS. These results show a much lower number of persons living with MS than expected, according to two recently published epidemiological studies estimating prevalence of MS in Hungary, using different methods.

The more recent study [31] used the 10th edition of the International Classification of Disease (ICD-10) and estimated the prevalence to be 130.8/100000. According to that estimate approximately 13000 persons with MS should live in Hungary, which would mean that almost 6000 people are "lost in the system". However, in Hungary the ICD-10 system does not include itemized settlements of accounts, meaning the code G35H0 could stand for several diagnoses in diseases that cause demyelination of the CNS. Thus, this method could have probably resulted in an overestimate of the prevalence of MS. Furthermore, this technique might only be helpful in sorting out people diagnosed with MS, other characteristics of the disease, such as disease course, DMT use, EDSS scores, psychopathology are not accessible, concluded by other authors [32].

The other recent epidemiological study [33] was based on rigorous regional registry data of Csongrád-county [34], where 4% of the Hungarian population lives, thus it can be considered representative. According to that paper the standardised prevalence of MS was 101.8/100.000, meaning that approximately 10000 persons with MS should live in the country. This method is considered more reliable, since registries provide extensive real-world, up-to-date data regarding information on people living with MS, formerly used in countless well-known epidemiological studies [35–41]. However, even when comparing the current study's number of persons living with MS to this estimate, a huge gap was discovered, meaning approximately 2500 people with MS are not receiving appropriate, MS specific medical care.

As previously mentioned, 2 care units did not participate in our assessment, and 2 participating centres did not provide information regarding number of persons living with MS, thus contributing to low number of people with MS. Another possible explanation behind this phenomenon could be that persons with MS after disease onset might not receive an accurate diagnosis, since misunderstanding and misapplication of the latest diagnostic criteria might still cause diagnostic problems among general neurologists and even among MS specialists [42,43], or if diagnosed with clinically isolated syndrome or RRMS correctly, people with MS might not attend doctor's appointments due to low disease burden or because of the lack of awareness of illness. Another reason could be that people with progressive phenotype are often undertreated. On one hand, persons with PPMS usually present with initial symptoms involving the lower limb, such as paraparesis or weakness, thus being misdiagnosed as disc hernia or other rheumatoid or orthopaedic disorder of the lower extremity [44]. However, diagnosing people with PPMS in time should be prioritized, since there is a therapeutic option that slows disease progression significantly [23], yet it can only be prescribed for people with PPMS having an EDSS score below 5.5. This finding suggests that educating GPs and other associate specialties to recognise PPMS could resolve this problem. On the other hand, severely disabled persons living with SPMS or PPMS no longer attend doctor's appointments because of their difficulty in mobility. Immobility is mainly caused by worsening spasticity and chronic pain thus appropriate management of these symptoms would be particularly important. However, according to our findings, spasticity and pain specialists are the least common neighbouring specialties in Hungarian MS care units, possibly contributing to low number of people living with MS and resulting in inequality in access to treatment.

Strengths of this study include, that this is the first assessment regarding the operation of MS care units in clinical practice, using the latest guidelines. Because of the high respondence rate this assessment can be considered representative, as it covers 93.55% of the Hungarian MS

care units. Due to this assessment, we identified the aspects that should be improved upon to provide better healthcare and possibly better quality of life for people with MS, long term.

Limitation of this study is that the results of the self-report questionnaires only represent whether MS centres fulfil MS care unit's minimum and recommended criteria. Yet it was not compared to patient reported outcome measures regarding satisfaction with quality of healthcare, since organising such an investigation is time and human resource consuming. The results could have been compared to patient information if Hungary owned a national multiple sclerosis registry. In addition to the absence of a national registry, 4 care units did not provide information regarding number of persons with MS, thus actual number of Hungarian people with MS could not have been determined, which also underlines the importance of creating and using a national MS registry. Another limitation of this survey is that it was conducted only in Hungary, thus it might seem that strictly national MS care units benefit from these results. Yet it could be the beginning of a survey series taking place in other countries, since being aware of the areas that need improvement would be beneficial for other health care system providers and receivers as well. In fact, this Hungarian survey is part of a bigger, more representative ongoing assessment in contribution with DSNS member countries.

## 5. Conclusion

In conclusion, according to our assessment to ensure equality to access treatment, more than half of the Hungarian MS care units might need improvements regarding personnel background and DMT use to meet all standards of the international guideline's recommendation and to provide most professional care for persons living with MS. Furthermore, the Hungarian population of people with MS receiving medical care in MS care units is below estimated, because people with primary progressive phenotype are often misdiagnosed or undiagnosed. Thus, presumably educating GPs and neighbouring specialties might improve diagnosis of PPMS, and the symptomatic treatment of severely disabled persons with progressive phenotype may also improve number of people with MS. At the same time, to be able to establish actual number of persons living with MS, a national MS registry is still a necessity.

## Acknowledgments

The authors thank the Danube Symposium for Neurological Sciences for collaboration.

The authors thank the Doctoral School of Clinical Medicine University of Szeged, Clinical and Experimental Neurosciences PhD program.

## Author Contributions

**Conceptualization:** László Vécsei, Krisztina Bencsik.

**Data curation:** Zsófia Kokas.

**Formal analysis:** Zsófia Kokas.

**Methodology:** László Vécsei, Krisztina Bencsik.

**Project administration:** Zsófia Kokas, Dániel Sandi.

**Resources:** Ágnes Köves, Ferenc Fazekas, Adrienne Jóri Birkás, Gabriella Katona, Krisztina Kovács, Dániel Milanovich, Enikő Dobos, István Kapás, Gábor Jakab, Tünde Csépány, Erzsébet Bense, Klotild Mátyás, Gábor Rum, Zoltán Szolnoki, István Deme, Zita Jobbágy, Dávid Kriston, Zsuzsanna Gerócs, Péter Diószeghy, László Bors, Adrián Varga, Levente Kerényi, Gabriella Molnár, Piroska Kristóf, Zsuzsanna Ágnes Nagy, Mária Sátori, Piroska Imre, Szilvia Péntek.

**Supervision:** Krisztina Bencsik.

**Visualization:** Zsófia Kokas, Dániel Sandi.

**Writing – original draft:** Zsófia Kokas.

**Writing – review & editing:** Dániel Sandi, Zsanett Fricska-Nagy, Judit Füvesi, Tamás Biernacki, Ágnes Köves, Ferenc Fazekas, Adrienne Jóri Birkás, Gabriella Katona, Krisztina Kovács, Dániel Milanovich, Enikő Dobos, István Kapás, Gábor Jakab, Tünde Csépány, Erzsébet Bense, Klotild Mátyás, Gábor Rum, Zoltán Szolnoki, István Deme, Zita Jobbágy, Dávid Kriston, Zsuzsanna Gerócs, Péter Diószeghy, László Bors, Adrián Varga, Levente Kerényi, Gabriella Molnár, Piroska Kristóf, Zsuzsanna Ágnes Nagy, Mária Sátori, Piroska Imre, Szilvia Péntek, Péter Klivényi, Zsigmond Tamás Kincses, László Vécsei, Krisztina Bencsik.

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
