## [Decision Letter · Decision Letter 0]

17 Nov 2021

PONE-D-21-33594Do Hungarian multiple sclerosis care units fulfil international criteria?PLOS ONE

Dear Dr. Bencsik,

Thank you for submitting your manuscript to PLOS ONE. After careful consideration, we feel that it has merit but does not fully meet PLOS ONE’s publication criteria as it currently stands. Therefore, we invite you to submit a revised version of the manuscript that addresses the points raised during the review process. Please submit your revised manuscript by Jan 01 2022 11:59PM. If you will need more time than this to complete your revisions, please reply to this message or contact the journal office at plosone@plos.org. Please include the following items when submitting your revised manuscript:A rebuttal letter that responds to each point raised by the academic editor and reviewer(s). You should upload this letter as a separate file labeled 'Response to Reviewers'.A marked-up copy of your manuscript that highlights changes made to the original version. You should upload this as a separate file labeled 'Revised Manuscript with Track Changes'.An unmarked version of your revised paper without tracked changes. You should upload this as a separate file labeled 'Manuscript'.

We look forward to receiving your revised manuscript.

Kind regards,

Marcello Moccia

Academic Editor

PLOS ONE

Journal Requirements:

"The current work was supported by GINOP 2.3.2-15-2016-00034, TUDFO/47138-1/2019-ITM and EFOP 3.6.3-VEKOP-16-2017-00009 project complementary scholarship for PhD students."

"KZs was supported by University of Szeged Open Access Fund, grant number 5535.

The Funder had no role in study design, data collection or analysis or preparation of the manuscript."

5. We note you have included a table to which you do not refer in the text of your manuscript. Please ensure that you refer to Tables 1, 2 and 5 in your text; if accepted, production will need this reference to link the reader to the Table.

Additional Editor Comments:

Authors have raised some minor comments I would like you to address in the revision.

Reviewers' comments:

Reviewer's Responses to Questions

**Comments to the Author**

1. Is the manuscript technically sound, and do the data support the conclusions?

Reviewer #1: Yes

Reviewer #2: Yes

2. Has the statistical analysis been performed appropriately and rigorously? 

Reviewer #1: N/A

Reviewer #2: Yes

3. Have the authors made all data underlying the findings in their manuscript fully available?

Reviewer #1: Yes

Reviewer #2: Yes

4. Is the manuscript presented in an intelligible fashion and written in standard English?

Reviewer #1: Yes

Reviewer #2: Yes

5. Review Comments to the Author

Reviewer #1: • Figura 2: Line 230-240

"X": It should probably be the opposite, as an X or + means usually that the criteria is fulfilled

Non-fullfilled could be blank (or grey) or -

It would be more easily understandable (as X usually in this type of graphic means a positive answer)

• Table 6. Lines 253-256

Again here X probably should be changed by grey (as it is) or - and delete X

• Lines 265-270

Please provide apart from %, the number of patients in each case before (% in brackets)

• Lines 271-278

Same commentary than previous one - n(%)

• Discussion Lines 285-403

Probably the following article, at least should be mentioned in the discussion (and in the literature review), as I believe, is the only one that has treated the same topic, from other points of view, but complementary to this article.:

Cristiano E, Abad P, Becker J, Carrá A, Correale J, Flores J, Fruns M, Garcea O, Garcia Bónitto J, Gracia F, Hamuy F, Navas C, Patrucco L, Rivera V, Velazquez M, Rojas JI. Multiple sclerosis care units in Latin America: Consensus recommendations about its objectives and functioning implementation. J Neurol Sci. 2021 Oct 15;429:118072. doi: 10.1016/j.jns.2021.118072. Epub 2021 Sep 8. PMID: 34509134.

Reviewer #2: The present study’s objective was to assess whether Hungarian MS care units fulfil international criteria.

Overall the methodology and statistics use is appropriate

I would suggest to use disease modifying therapy instead of immunomodulatory therapies.

As well persons or people with MS instead of MS patients would be more appropriate term.

Authors should explain why only data from Hungary are reported, as they stated that “Danube Symposium for Neurological Sciences member countries decided to create the Multiple Sclerosis National Symposium, aiming to collect and assess data regarding inpatient and outpatient care of people with MS.”

6. PLOS authors have the option to publish the peer review history of their article (what does this mean?). If published, this will include your full peer review and any attached files.

Reviewer #1: No

Reviewer #2: No

---

## [Author Response · Author response to Decision Letter 0]

21 Jan 2022

We are grateful for the quick response and the helpful recommendations.

Review comments made by the Reviewers:

Reviewer #1

1) Figure 2, lines 230-240

Figure 2 and legend of Figure 2 were changed according to the Reviewer’s observation, from:

“X indicates unfulfilled criterion, + indicates that all minimum and/or recommended criteria are met”

to:

“X indicates that criterion is met”

2) Table 6, lines 253-256

Table 6 and legend of Table 6 were altered pursuant to the Reviewer’s comment, from:

“� indicates that the disease modifying therapy is used in the care units, X with a grey background indicates that the disease modifying therapy is not used in the care unit, N/A with a grey background means that no data were supplied”

to:

“X indicates that the disease modifying therapy is used in the care units, cell with a grey background indicates that the disease modifying therapy is not used in the care unit, “N/A” means that no data were supplied.

3) Lines 265-270

Number of patients were added in each case, and percentages were placed into round brackets according to the Reviewer’s recommendation.

4) Lines 271-278

Number of patients were added in each case, and percentages were placed into round brackets in line with the Reviewer’s suggestion.

5) Discussion lines 285-403

The reference recommended by the Reviewer was added as a new paragraph to the Discussion Section:

“As previously mentioned, no surveys were conducted to assess whether or how MS care unit criteria are fulfilled in real-world practice. However, these criteria were carefully reviewed from a different aspect by a panel of neurologists from Latin America [20]. In Latin America diagnosis and treatment of MS was usually carried out by general neurologists, and care of people with MS was not perused in specialized centres, thus the panel’s objective was to create a realistic adaptation of the MS care unit recommendations considering regional differences regarding healthcare in Latin American countries. Cristiano et al acknowledged the importance of MSCUs to optimize MS care and reached consensus on what a care unit in Latin America should offer. However, they also concluded, that despite the desire to meet the personnel and instrumental requirements of an MSCU, it would be difficult to implement these expectations in certain regions of Latin America. Thus, they determined a standardized protocol for MRI assessments and recommended education of general neurologists to facilitate accurate diagnosis of MS. At the same time, they also proposed the idea of close collaboration between general neurologists and MSCUs, and similarly to our assessment they propounded the feasibility of reference care units.”

The reference recommended by the Reviewer was also added to the Reference Section:

“20. Cristiano E, Abad P, Becker J, Carrá A, Correale JF, Fruns M, et al. Multiple clerosis care in Latin America: Consensus recommendations about its objectives and functioning implemetation. J Neurol. Sci. 2021 Oct 14;429:118072. https://doi.org/10.1016/j.jns.2021.118072”

Reviewer #2

1) The expression “immune modulatory therapy” was switched to “disease modifying therapy” in the Introduction Section, according to the Reviewer’s advice.

2) The term “MS patients” was altered to either “persons with MS”, “persons living with MS” or “people with MS”, “people living with MS” throughout the manuscript, in line with the Reviewer’s proposition.

3) Explanation of “why only data from Hungary were reported” suggested by the Reviewer, was added at the end of the Introduction Section’s Objectives Subheading:

“The present study is part of an international survey series conducted in DSNS member countries. Taking into account that such robust data collection is time and human resource demanding, data acquisition and analysis from DSNS member countries is in process. At the same time, considering the lack of similar assessments in the past, the present study can be considered as a pilot investigation aiming to collect data only from Hungarian MS care units, before presenting international outcomes.”

---

## [Decision Letter · Decision Letter 1]

9 Feb 2022

Do Hungarian multiple sclerosis care units fulfil international criteria?

PONE-D-21-33594R1

Dear Dr. Bencsik,

We’re pleased to inform you that your manuscript has been judged scientifically suitable for publication and will be formally accepted for publication once it meets all outstanding technical requirements.

Kind regards,

Marcello Moccia

Academic Editor

PLOS ONE

Additional Editor Comments (optional):

Reviewers' comments:

Reviewer's Responses to Questions

**Comments to the Author**

1. If the authors have adequately addressed your comments raised in a previous round of review and you feel that this manuscript is now acceptable for publication, you may indicate that here to bypass the “Comments to the Author” section, enter your conflict of interest statement in the “Confidential to Editor” section, and submit your "Accept" recommendation.

Reviewer #1: (No Response)

Reviewer #2: All comments have been addressed

2. Is the manuscript technically sound, and do the data support the conclusions?

Reviewer #1: Yes

Reviewer #2: Yes

3. Has the statistical analysis been performed appropriately and rigorously? 

Reviewer #1: Yes

Reviewer #2: Yes

4. Have the authors made all data underlying the findings in their manuscript fully available?

Reviewer #1: Yes

Reviewer #2: Yes

5. Is the manuscript presented in an intelligible fashion and written in standard English?

Reviewer #1: Yes

Reviewer #2: Yes

6. Review Comments to the Author

Reviewer #1: Now the paper is very well adapted, BUT there are docenzs of "persons with multiple sclerosis". I really believe they could be replaced by PwMS, it will make the paper much more readable. No other concerns

Reviewer #2: (No Response)

7. PLOS authors have the option to publish the peer review history of their article (what does this mean?). If published, this will include your full peer review and any attached files.

Reviewer #1: No

Reviewer #2: **Yes: **Mario Habek

---

## [Editor Report · Acceptance letter]

23 Feb 2022

PONE-D-21-33594R1 

Do Hungarian multiple sclerosis care units fulfil international criteria? 

Dear Dr. Bencsik:

I'm pleased to inform you that your manuscript has been deemed suitable for publication in PLOS ONE. Congratulations! Your manuscript is now with our production department. 

Kind regards, 

on behalf of

Dr. Marcello Moccia 

Academic Editor

PLOS ONE